# Chronic Kidney Disease Has No Impact on Tear Film Substance P Concentration in Type 2 Diabetes

**DOI:** 10.3390/biomedicines11092368

**Published:** 2023-08-24

**Authors:** Kofi Asiedu, Sultan Alotaibi, Arun V. Krishnan, Natalie Kwai, Ann Poynten, Maria Markoulli, Roshan Dhanapalaratnam

**Affiliations:** 1School of Optometry & Vision Science, University of New South Wales, Sydney, NSW 2052, Australia; 2Department of Optometry and Vision Science, College of Applied Medical Science, King Saud University, Riyadh 11421, Saudi Arabia; 3School of Clinical Medicine, University of New South Wales, Sydney, NSW 2052, Australia; 4School of Medical, Indigenous and Health Sciences, Faculty of Science, Medicine and Health, University of Wollongong, Wollongong, NSW 2522, Australia; 5Department of Endocrinology, Prince of Wales Hospital, Sydney, NSW 2031, Australia

**Keywords:** diabetic neuropathy, peripheral neuropathy, substance p, chronic kidney disease, cornea

## Abstract

Purpose: The study aimed to ascertain the potential effects of chronic kidney disease (CKD) on substance P concentration in the tear film of people with type 2 diabetes. Methods: Participants were classified into two groups: type 2 diabetes with concurrent chronic kidney disease (T2DM–CKD (n = 25)) and type 2 diabetes without chronic kidney disease (T2DM–no CKD (n = 25)). Ocular surface discomfort assessment, flush tear collection, in-vivo corneal confocal microscopy, and peripheral neuropathy assessment were conducted. Enzyme-linked immunosorbent assays were utilized to ascertain the levels of tear film substance P in collected flush tears. Correlation analysis, hierarchical multiple linear regression analysis, and t-tests or Mann–Whitney U tests were used in the analysis of data for two-group comparisons. Results: There was no substantial difference between the T2DM–CKD and T2DM–no CKD groups for tear film substance P concentration (4.4 (0.2–50.4) and 5.9 (0.2–47.2) ng/mL, respectively; p = 0.54). No difference was observed in tear film substance P concentration between the low-severity peripheral neuropathy and high-severity peripheral neuropathy groups (4.4 (0.2–50.4) and 3.3 (0.3–40.7) ng/mL, respectively; p = 0.80). Corneal nerve fiber length (9.8 ± 4.6 and 12.4 ± 3.8 mm/mm^2^, respectively; p = 0.04) and corneal nerve fiber density (14.7 ± 8.5 and 21.1 ± 7.0 no/mm^2^, respectively; p < 0.01) were reduced significantly in the T2DM–CKD group compared to the T2DM–no CKD group. There were significant differences in corneal nerve fiber density (21.0 ± 8.1 and 15.8 ± 7.7 no/mm^2^, respectively; p = 0.04) and corneal nerve fiber length (12.9 ± 4.2 and 9.7 ± 3.8 mm/mm^2^, respectively; p = 0.03) between the low- and high-severity peripheral neuropathy groups. Conclusion: In conclusion, no significant difference in tear film substance P concentration was observed between type 2 diabetes with and without CKD. Corneal nerve loss, however, was more significant in type 2 diabetes with chronic kidney disease compared to type 2 diabetes alone, indicating that corneal nerve morphological measures could serve greater utility as a tool to detect neuropathy and nephropathy-related corneal nerve changes.

## 1. Introduction

Peripheral neuropathy affects 78% of people with pre-dialysis chronic kidney disease [1]. Peripheral neuropathy occurs in 50% of individuals with type 2 diabetes, leading to abnormal sensation, foot ulceration, and, in severe and untreated cases, limb amputation [2]. In people with chronic kidney disease resulting from diabetes, the combination of these two metabolic diseases produces a variant of peripheral neuropathy that is more severe and faster progressing than either disease alone [1]. Hence, there is a need to explore and find more sensitive biomarkers to pick up early signs of neuropathy in diabetic chronic kidney disease.

Recent studies have shown greater corneal nerve loss in diabetic chronic kidney disease compared to type 2 diabetes alone when the severity of peripheral neuropathy is matched between the two groups [3]. Tear film substance P is mainly expressed and released by corneal neurons and exerts its physiological effects through the neurokinin receptors on the ocular surface [4,5]. Hence, the idea or hypothesis explored in the present study is that loss of corneal nerve fibers because of diabetic chronic kidney disease [3] may be associated with an alteration in the concentrations of substance P in tears. Measuring tear film neuropeptides such as substance P concentration may assist in detecting subclinical neuropathy in diabetic chronic kidney disease.

Substance P is involved in neuro-immunoregulation and the maintenance of ocular surface comfort and homeostasis [4,5]. Substance P is known to induce pro-inflammatory responses through its receptors on the immune cells in various ocular surface pathologies [6]. Substance P is critical in neurogenic inflammation and corneal epithelial wound healing, and it is reduced in conditions like diabetic neuropathy in type 1 diabetes [6,7]. Substance P induces the generation and release of chemokines and neurokines which stimulate immune cell recruitment, worsening inflammation [6]. Studies in the past have shown that substance P significantly increases the release of chemotactic cytokines such as IL-8, which recruit immune cells such as lymphocytes to the site of inflammation on the ocular surface [6,8]. Ocular surface neuroinflammatory status is implicated in the pathogenesis of both ocular surface discomfort and dry eye disease [9,10]. This suggests that tear film substance P concentration may be an indicator of tear film neuroinflammatory status.

In vivo corneal confocal microscopy, which is used to evaluate corneal nerves, is limited because only a small fraction of the total ocular surface area is imaged at a single point in time [7]. This represents a small area of view for consistent, systematic, and precise quantification of the total corneal surface nerves. Corneal confocal microscopy cannot image corneal epithelial nerve endings, which may be affected early in the disease [7]. Since substance P is secreted from corneal nerve endings, it is plausible that measuring tear film substance P concentration may reveal early or subtle changes in the corneal nerve endings beyond the capability of the in vivo corneal confocal microscope.

Owing to the greater corneal neuropathy occurring in diabetic chronic kidney disease compared to type 2 diabetes alone [3], as well as the potential role of substance P in kidney function [11,12], we postulate that there may be differences in the tear film substance P concentration between individuals with type 2 diabetes with chronic kidney disease compared those with only type 2 diabetes.

## 2. Methods

This was a prospective, cross-sectional study involving 50 people with type 2 diabetes (25 people with T2DM-CKD and 25 people with T2DM–no CKD) recruited consecutively from the Diabetes Centre at the Prince of Wales Hospital, Sydney, Australia. Each participant provided written informed consent. Only participants meeting the following criteria were included in the study:All participants recruited into the study were above 18 years of age and gave written informed consent.Recruitment was restricted to persons with type 2 diabetes.

Participants were excluded if they had a history of any disease known to cause neuropathy such as the use of chemotherapy or immunosuppressive medications and vitamin B12 deficiency, corneal abrasion, allergies to anesthetic eye drops, usage of any topical steroidal and non-steroidal anti-inflammatories, corneal ectasia, current ocular infection, were on any contact lens modality, or had undergone cataract surgery in the last half of the year. The study received ethical approval from the University of New South Wales and the Research Ethics Committee of the South Eastern Sydney Local Health District. The protocols, tests, and investigations conducted in the study were performed according to the tenets of the Declaration of Helsinki (2013). 

### 2.1. Ocular Surface Assessment

Ocular surface discomfort and ocular pain were evaluated with the aid of the validated Ocular Surface Disease Index and Ocular Pain Assessment Survey, respectively [13,14]. The summary scores of both questionnaires were used in a subsequent analysis of ocular surface discomfort and ocular pain. A drop of normal saline was used to wet the fluorescein-impregnated paper strip and the paper strip was then gently applied to the temporal side of the conjunctiva while the individual looked nasally. The ocular surface was examined with a slit lamp biomicroscope with a yellow barrier filter using cobalt blue filtered light. The cornea and conjunctiva were assessed for any epithelial defects with the Oxford grading scale (0 to 15) [15]. The yellow barrier filter was employed to allow conjunctival staining to be seen alongside the corneal staining [16]. This is shown in Table 1.

### 2.2. Sample Size Calculation

With the aid of G*Power 3.1.9.4 (Heinrich Heine University, Dusseldorf, Germany), the estimated sample size was based on a difference between mean values of tear film substance P concentration in people with diabetes (1472.9 ± 1670.9 P pg/mL) compared with healthy controls (4150.2 ± 4752.0 pg/mL) [17]. In each group, a minimum of 23 participants is adequate and required to show a significant difference with 80% power at an α level of 0.05.

### 2.3. Indicators of Kidney Function

In the present study, chronic kidney disease was considered as an estimated glomerular filtration rate (eGFR) of less than 60 mL/min/1.73 m^2^, combined with concurrent mild to severe albuminuria as stipulated by the Kidney Disease: Improving Global Outcomes Guideline [18]. All participants with type 2 diabetes with concurrent chronic kidney disease had raised urea and creatinine levels in serum, as shown in Table 1. All participants were grouped according to their nephropathy status, namely, type 2 diabetes without chronic kidney disease (T2DM-no CKD) and type 2 diabetes with chronic kidney disease (T2DM-CKD). Body mass index, serum potassium, urea, creatinine, total cholesterol, high-density lipoprotein, low-density lipoprotein, triglycerides, and HbA1c were collected during the clinical assessment of the participants and extracted from the electronic medical records, having been performed within the past 4 weeks.

### 2.4. Corneal Confocal Microscopy

Corneal confocal microscopy (Heidelberg Retinal Tomograph III Rostock Cornea Module; Heidelberg Engineering GmbH, Heidelberg, Germany) was used to image the right eye only considering that corneal nerve morphology is symmetrical between eyes [19,20]. Oxybuprocaine hydrochloride 0.4% (Bausch & Lomb, Chatswood, Australia) was topically applied to anesthetize both eyes. An optimal quantity of 2.5% hydroxypropyl methylcellulose (GenTeal gel, Alcon Inc., Worth Fort, TX, USA) was instilled onto the imaging lens and a new Tomocap (Heidelberg Retinal Tomograph III Rostock Cornea Module; Heidelberg Engineering GmbH, Heidelberg, Germany) was carefully fitted onto the imaging probe.

Images were taken from the cornea at the right eye’s central and inferior whorl regions. Four inferior whorl and eight central images not overlapping by more than 20% for each person were selected and used in the corneal nerve parameter analysis [21]. Images were analyzed using automated nerve analysis software (Corneal Nerve Fiber Analyzer V.2, ACCMetrics, University of Manchester, Manchester, UK) [22]. The customized software calculated the corneal nerve fiber length (the total length of major nerves and their branches per square millimeter), corneal nerve fiber density (the exact number or counts of main nerves per square millimeter), and inferior whorl length (the nerve fiber length at the corneal inferior whorl region).

### 2.5. Tear Sample Collection

A micropipette was used to instill 20 μL normal saline (sodium chloride injection 0.9%; Pfizer, Sydney, Australia) into the inferior cul-de-sac of the lower eyelid of both eyes. Participants were instructed to tilt their heads toward the side of the collection and a 10 μL glass microcapillary tube was used to collect tears from the temporal canthus (Blaubrand intraMark, Wertheim, Germany) for 1 min. The 1 min time of collection helped minimize reflex tearing [17]. The tears collected were drained into an Eppendorf tube. The collected tears were centrifuged at 4000 rpm for 20 min at 4 °C to allow cellular debris to settle to the bottom. Tear samples were then stored as two 5 μL aliquots at −80 °C before analysis. The tear flow rate was estimated for all tear samples collected by recording the time and volume of tears.

### 2.6. ELISA of Substance P Concentration

The substance P concentration in the flush tear samples was assessed using enzyme-linked immunosorbent assays, adhering to the protocol and procedures stipulated by the manufacturer (Cayman Chemical Company, Ann Arbor, MI, USA). The dilution factor employed in the analysis was 1:30. Tear samples for each participant were analyzed in duplicate. The samples were in duplicate along with an alkaline phosphate-conjugated substance P antigen and a polyclonal rabbit antibody specific to substance P. The optical density was taken at 595 nm on a microplate reader using the Omega series reader software (BMG LABTECH, Victoria, Australia), and a (%Bound/Maximum Bound) standard curve was used to compute individual concentrations.

### 2.7. Peripheral Neuropathy Assessment and Diagnosis

The Total Neuropathy Score was used to evaluate the severity of peripheral neuropathy in both groups [23]. The Total Neuropathy Score has been validated in both type 2 diabetes and chronic kidney disease [1,23]. The assessment is completed across eight domains with each domain having a scoring ranging from 0 to 4, with 0 implying no abnormality and 4 representing the severest abnormality. Scores from each domain are added up to constitute the Total Neuropathy Score ranging from 0 and 32 [23]. Participants with a Total Neuropathy Score between 0 and 8 were considered as having low-severity peripheral neuropathy and those with Total Neuropathy Score greater than 8 were classified as having high-severity peripheral neuropathy [1,24].

### 2.8. Data Analysis

Data analyses were conducted using SPSS version 23 (IBM Corp: Armonk, NY, USA) and Graph Pad Prism 9.0 (Graph Pad Software Inc., San Diego, CA, USA). Descriptive statistics were computed as percentages and counts for categorical variables and as means and standard deviations for continuous data. All the variables were continuous variables, apart from sex, which was categorical, so the χ2 test was used. Normality testing was performed with the Shapiro–Wilk test and the visual assessment of the quantile–quantile and detrended plots. Data that showed normal distribution, such as the corneal nerve parameters, were analyzed using an independent sample t-test. For nonparametric data, such as tear film substance P concentration, a Mann–Whitney U test was used to determine any substantial differences between the two diabetes groups. A *p* < 0.05 was considered statistically significant. Pearson r correlation analysis was performed to determine the relationship between tear film substance P concentration (logarithmically transformed) and ocular surface parameters as well as metabolic indicators. Finally, hierarchical multiple linear regression analysis controlling for age, duration of diagnosis, and Total Neuropathy Score was conducted to ascertain whether corneal nerve morphological measures and kidney function tests explained any significant variance in the logarithmically transformed tear film substance P concentration. Hierarchical multiple linear regression was conducted to control for the potential effects of the duration of disease as the means of the two groups were quite different even though not reaching statistical significance.

## 3. Results

### 3.1. Participant Demographic Data and Metabolic Parameters

Participant characteristics, including demographic data, Total Neuropathy Scores, and metabolic parameters, are documented in Table 1. Participants with T2DM–no CKD and T2DM–CKD were matched for age, Total Neuropathy Score, body mass index (BMI), sex distribution, and duration of diabetes (Table 1). There was a trend of a longer duration of type 2 diabetes in the T2DM–CKD compared to T2DM–no CKD; however, this difference did not reach statistical significance (Table 1). Participants were equivalent in terms of total serum cholesterol, low-density lipoprotein, serum triglycerides, high-density lipoprotein, and HbA1c. Serum urea levels, potassium, urinary albumin/creatinine ratio, and creatinine were higher in the T2DM–CKD group than in the T2DM–no CKD group. Participants with a diagnosis of peripheral neuropathy based on the Total Neuropathy Score were categorized into low- and high-severity peripheral neuropathy groups. They were matched for age, sex distribution, duration of diagnosis, and eGFR, as shown in Table 2.

### 3.2. Tear Film Substance P Concentration

The concentration of substance P in the tear film was not statistically different between T2DM–CKD and T2DM–no CKD (4.4 (0.2–50.4) and 5.9 (0.2–47.2) ng/mL, respectively; p = 0.54). Similarly, when participants with low-severity peripheral neuropathy were compared to participants with high-severity peripheral neuropathy, no statistically significant differences in tear film substance P concentration were observed between the groups (4.4 (0.2–50.4) and 3.3 (0.3–40.7) ng/mL, respectively; p = 0.80).

### 3.3. Corneal Nerve Morphological Parameters

Corneal nerve data analysis in the T2DM–CKD group compared with the T2DM–no CKD group showed reduced corneal nerve fiber length (9.8 ± 4.6 and 12.4 ± 3.8 mm/mm^2^, respectively; p = 0.04) and corneal nerve fiber density (14.7 ± 8.5 and 21.1 ± 7.0 no/mm^2^, respectively; p < 0.01). Inferior whorl length was not significantly different in the T2DM–no CKD group compared to the T2DM–CKD (9.7 ± 4.6 and 8.1 ± 4.0, respectively, p = 0.21, Table 3, Figure 1).

There were significant differences in corneal nerve fiber density (21.0 ± 8.1 and 15.8 ± 7.7 no/mm^2^, respectively; p = 0.04) and corneal nerve fiber length (12.9 ± 4.2 and 9.7 ± 3.8 mm/mm^2^, respectively; p = 0.03) between the low- and high-severity peripheral neuropathy groups as shown in Table 2.

### 3.4. Correlational and Hierarchical Multiple Regression Analysis

There was no significant correlational relationship between tear film substance P concentration (logarithmic transformation) and the following parameters: corneal nerve fiber density (r = −0.13, p = 0.39), corneal nerve fiber length (r = −0.16, p = 0.27), inferior whorl length, r = −0.02, p = 0.90), estimated glomerular filtration rate (r = −0.01, p = 0.93), and urinary albumin creatinine ratio (r = −0.03, p = 0.84). This finding was confirmed in the hierarchical multiple linear regression analysis (which was adjusted for age, duration diagnosis, and Total Neuropathy Score), which demonstrated that corneal nerve parameters and measures of kidney function did not explain the variances in tear film substance P concentration observed in the sample, as shown in Table 4.

## 4. Discussion

The current study sought to ascertain whether changes in corneal nerve parameters are associated with an alteration in tear film substance P concentration in people with type 2 diabetes with concurrent chronic kidney disease compared to people with type 2 diabetes alone. Even though we demonstrated that corneal nerve parameters were reduced in type 2 diabetes with concurrent chronic kidney disease, consistent with previous findings [3], we found no difference in tear film substance P concentration between the two diabetes groups.

Longstanding hyperglycemia in type 2 diabetes is a major cause of corneal nerve loss and peripheral nerve injury, resulting from advanced glycation end products, disruptions of axonal function, uncontrolled oxidative stress and reactive oxygen species, and abnormal hemodynamic regulatory pathways [25]. Concurrent chronic kidney disease in type 2 diabetes worsens these abnormalities [26,27], facilitating a more severe corneal nerve loss. Furthermore, chronic kidney disease is closely associated with hyperkalemia, which may mediate nerve dysfunction in chronic kidney disease by altering nerve ion channel function [28]. Nerve dysfunction is strongly associated with the reduction in sodium–potassium pump activity, which has been shown in type 2 diabetes [29]. Reduced sodium–potassium pump activity and hyperkalemia may combine synergistically to impair corneal nerves and other peripheral nerves more significantly in diabetic chronic kidney disease more so than in type 2 diabetes alone.

Dry eye disease is of high prevalence in persons with type 2 diabetes [30]. Ocular surface inflammation plays a huge role in the pathogenesis of dry eye disease and ocular surface discomfort [31]. Inflammation has a significant impact on how dry eye disease in people with type 2 diabetes develops [31]. In laser-assisted in situ keratomileusis (LASIK), corneal nerve loss is accompanied by more severe ocular surface discomfort [32]; however, loss of corneal nerve fibers in LASIK is distinctively different from type 2 diabetes. In LASIK, the corneal nerve loss is rapid, resulting in an increased level of tear film substance P concentration [33], which enhances neuroinflammation; however, in type 2 diabetes, the loss of corneal nerve fibers is gradual, and no difference in tear film substance P concentration is observed between persons with type 2 diabetes and healthy controls [7,34]. The presence of chronic kidney disease may lead to more severe and faster progression of neuropathy, implying that additional corneal nerve loss in diabetic chronic kidney disease may alter tear film substance P concentration. However, no differences in tear film substance P concentration were observed between the two groups, which is consistent with the finding that there were no significant differences in ocular surface discomfort and ocular pain between the two groups in the current study.

Possible explanations for this observation may also include differences in the pathophysiology of peripheral neuropathy occurring in type 2 diabetes versus chronic kidney disease. In contrast to somatic innervation, corneal innervation is devoid of A-beta fibers [35]. Sensory nerves within the cornea are composed of high-speed thinly myelinated Aδ fibers with a larger diameter, which constitute approximately 30% of corneal afferent fibers, and slower-conducting unmyelinated C-fibers with a small diameter, which make up 70% of corneal afferent fibers [35,36,37]. In rat dorsal root ganglion, substance P appears to be expressed mainly by C-fibers instead of the Aδ-fibers [38]. Other studies have shown that large-sized corneal trigeminal ganglion neurons do not normally express and release substance P [39]. Bae et al. reported that in rat trigeminal ganglion, 98% of cell bodies expressing substance P have a cross-sectional area < 800 μm [39]. Another study showed that corneal trigeminal neurons with a cross-sectional area > 800 μm^2^ do not express substance P in homeostasis [40]. The C-fibers in the cornea may be affected early in diabetes [41,42], causing a significant reduction in substance P concentration in tears before the onset of chronic kidney disease. Concurrent chronic kidney disease may occur in type 2 diabetes later, but at this time, significant C-fibers may have already been lost, hence potentially explaining why no further reduction in tear film substance P concentration was observed with the additional loss of corneal nerve fibers.

Previous reports suggest that pre-dialysis chronic kidney disease affects large nerve fibers without significant small nerve fiber involvement [43]. Hence, the presence of pre-dialysis chronic kidney disease in type 2 diabetes represents a condition that preferentially targets large nerve fibers without significant small nerve fiber involvement. There is some probability that the additional corneal nerve loss observed in people with type 2 diabetes with concurrent chronic kidney disease may reflect large diameter Aδ-fiber loss, which plays a minor role in substance P secretion. The findings in this study are similar to those in people with toxic neuropathy due to chemotherapy drugs such as oxaliplatin and paclitaxel [44]. Oxaliplatin affects predominantly large nerve fibers while paclitaxel affects both large and small nerve fibers [45]. Previous studies have shown that both paclitaxel and oxaliplatin are associated with corneal nerve fiber loss but only paclitaxel was associated with a reduction in tear film substance P concentration [44]. Our results, in conjunction with findings in patients treated with chemotherapeutic medications, suggest that tear film substance P concentration may only be reduced in forms of neuropathy where there is both small nerve fiber and large nerve fiber injury [44].

There are several limitations of the current study worth highlighting. The study did not include a matched healthy control group, implying that the study findings are limited to people with type 2 diabetes and not generalizable to other populations. Even though the minimum sample size required to demonstrate a significant difference was used, a larger sample size will improve the robustness of the study findings. Future cross-sectional and longitudinal studies utilizing larger sample sizes and a healthy control group may be needed to confirm the findings of the current study.

In conclusion, no significant difference in tear film substance P concentration was observed between type 2 diabetes with and without chronic kidney disease. Corneal nerve loss, however, was more significant in type 2 diabetes with concurrent chronic kidney disease compared to type 2 diabetes alone. This suggests that corneal nerve parameters may serve greater utility as it continues to be a promising tool in the detection of neuropathy in type 2 diabetes with concurrent chronic kidney disease compared to tear film substance P concentration measurement.

## Figures and Tables

**Figure 1 biomedicines-11-02368-f001:**
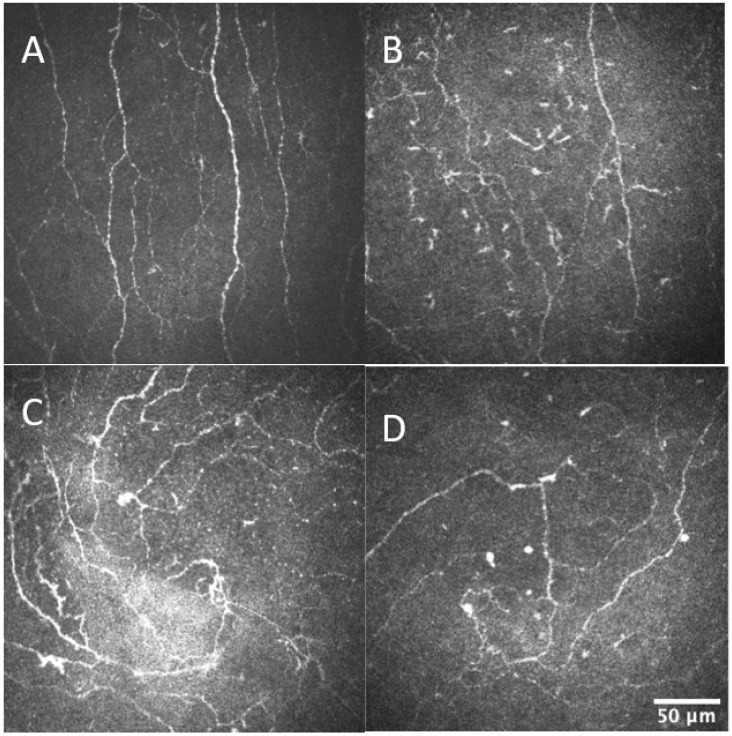
Showing central and inferior whorl images of T2DM-no CKD (**A**,**C**) and T2DM-CKD (**B**,**D**) demonstrating corneal nerve loss in T2DM-CKD compared to T2DM-no CKD.

**Table 1 biomedicines-11-02368-t001:** Demographic and clinical characteristics of the participants in the type 2 diabetes with or without chronic kidney disease groups; results are expressed as Mean ± SD.

Parameter	T2DM-CKD (n = 25)	T2DM-no CKD (n = 25)	p-Value
Age, years	70.8 ± 8.5	68.2 ± 8.5	p = 0.29
Sex, % male	64	72	p = 0.54
Body mass index, kg/m^2^	31.7 ± 6.9	32.1 ± 6.7	p = 0.98
Duration of diagnosis, years	20.7 ± 8.8	14.7 ± 12.5	p = 0.06
HbA1c, %	8.1 ± 1.8	8.7 ± 2.1	p = 0.25
Serum urea, mg/dL	10.8 ± 4.6	6.4 ± 1.9	p < 0.001
Creatinine, mg/dL	171.2 ±118.5	76.3 ± 16.1	p < 0.001
Estimated glomerular filtration rate, mL/min/1.73 m^2^	41.3 ± 19.2	80.4 ± 10.9	p < 0.001
Urine ACR, mg/mmol	44.3 ± 88.1	3.8 ±3.9	p = 0.03
Serum potassium, mmol/l	4.5 ± 0.3	4.3 ± 0.5	p = 0.04
Total cholesterol, mmol/L	3.8 ± 1.1	3.8 ± 1.0	p = 0.71
High-density lipoprotein, mmol/L	1.1 ± 0.4	1.3 ± 0.4	p = 0.24
Low-density lipoprotein, mmol/L	1.8 ± 0.8	1.7 ± 0.9	p = 0.80
Triglycerides, mmol/L	2.2 ± 2.1	1.7 ± 1.4	p = 0.40
Total Neuropathy Score (scores)	6.9 ± 5.4	6.4 ± 5.2	p = 0.74
Ocular surface staining (scores)	3.6 ± 1.5	2.7 ± 2.3	p = 0.09
Ocular Surface Disease Index (scores)	13.0 ± 11.4	12.6 ± 11.6	p = 0.76
Ocular Pain Assessment Survey (scores)	3.5 ± 5.6	3.6 ± 4.9	p = 0.96

**Table 2 biomedicines-11-02368-t002:** Demographic and clinical characteristics of the participants in the low and high peripheral neuropathy groups.

Parameter	High-Severity Neuropathy Group (TNS Grade 3–4) (n = 25)	Low-Severity Neuropathy Group (TNS Grade 0–2) (n = 25)	p-Value
Age, years	68.8 ± 10.9	69.3 ± 5.9	p = 0.86
Duration of disease, years	17.5 ± 11.0	15.0 ± 9.4	p = 0.61
Estimated glomerular filtration rate, mL/min/1.73 m^2^	59.5 ± 23.2	64.9 ± 21.3	p = 0.44
Corneal nerve fiber density (no./mm^2^)	15.8 ± 7.7	21.0 ± 8.1	p = 0.04
Corneal nerve fiber length (mm/mm^2^)	9.9 ± 4.0	12.9 ± 4.2	p = 0.03
Inferior whorl length (IWL) (mm/mm^2^)	8.6 ± 2.8	10.0 ± 5.0	p = 0.29
Substance P concentration (ng/mL)	3.3 (0.3–40.7) *	4.4 (0.2–50.4) *	p = 0.80
Ocular surface staining (scores)	3.6 ± 1.7	3.0 ± 2.1	p = 0.33
Ocular surface disease index (scores)	13.2 ± 13.0	9.2 ± 9.2	p = 0.21

* Inter-quantile range.

**Table 3 biomedicines-11-02368-t003:** Ocular surface parameters in type 2 diabetes with chronic kidney disease (T2DM-CKD) and type 2 diabetes without chronic kidney disease (T2DM-NO CKD).

Parameter	T2DM-CKD	T2DM-No CKD	p-Value
Corneal nerve fiber density (no./mm^2^)	14.7 ± 8.5	21.1 ± 7.0	p < 0.01
Corneal nerve fiber length (mm/mm^2^)	9.8 ± 4.6	12.4 ± 3.8	p = 0.04
Inferior whorl length (mm/mm^2^)	8.1 ± 4.0	9.7 ± 4.9	p = 0.21
Substance P (ng/mL)	4.4 (0.6–40.8) *	5.9 (0.2–47.2) *	p = 0.54

* Inter-quantile range.

**Table 4 biomedicines-11-02368-t004:** Hierarchical multiple linear regression analysis summarizing the associations between tear film substance P concentration and corneal nerve parameters and kidney function measures controlling for age, duration of diagnosis, and Total Neuropathy Score.

Parameter	β Coefficient	Standard Error	p-Value	95% Lower	95% Upper
Dependent Variable: Tear Film Substance P Concentration					
Corneal nerve fiber density (no./mm^2^)	0.04	0.06	0.42	−0.07	0.16
Corneal nerve fiber length (mm/mm^2^)	−0.11	0.10	0.28	−0.32	0.10
Inferior whorl length (mm/mm^2^)	0.02	0.04	0.63	−0.06	0.10
Estimated glomerular filtration rate (mL/min/1.73 m^2^)	−0.01	0.007	0.50	−0.02	0.01
Urinary albumin/ creatinine ratio (mg/mmol)	−0.001	0.002	0.69	−0.06	0.004

## Data Availability

The data presented in this study are available on reasonable request from the corresponding author. The data is not publicly available due to ethical reasons.

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
