# Peer review of "Chronic Kidney Disease Has No Impact on Tear Film Substance P Concentration in Type 2 Diabetes"

_biomedicines, 2023, doi:10.3390/biomedicines11092368_

Round 1
Reviewer 1 Report
Dry eye disease (DED) is very common in diabetic patients. The inflammation has a significant impact in the development of DED in diabetic patients. (Byambajav, M.; Collier, A.;Shu, X.; Hagan, S. Tear Fluid Biomarkers and Quality of Life in People with Type 2 Diabetes and Dry Eye Disease. Metabolites 2023, 13, 733. https://doi.org/10.3390/ metabo13060733)
SP is known to induce pro-inflammatory effects through autocrine or paracrine action on the immune cells in various ocular surface pathologies, including DED, as well. (Singh RB et al: Modulating the tachykinin: Role of substance P and neurokinin receptor expression in ocular surface disorders. Ocul Surf. 2022 July ; 25: 142–153. doi:10.1016/j.jtos.2022.06.007.)
Is there any data on whether the 2 study groups differ from each other and from age-matched healthy controls in terms of dry eye symptoms?
A possible explanation for the non-significant differences in tear film substance P concentration between the two study groups could be dry eye disease.
Author Response
Reviewer 1
1.Dry eye disease (DED) is very common in diabetic patients. The inflammation has a significant impact in the development of DED in diabetic patients. (Byambajav, M.; Collier, A.;Shu, X.; Hagan, S. Tear Fluid Biomarkers and Quality of Life in People with Type 2 Diabetes and Dry Eye Disease. Metabolites 2023, 13, 733. https://doi.org/10.3390/ metabo13060733)
Response
Thank you for the suggestion. We have included this statement in our discussion. This is on Line 286-289 which reads “Dry eye disease is very common in people with type 2 diabetes.41 Ocular surface inflammation is central to the pathogenesis of dry eye disease and ocular surface discomfort.42 Inflammation has a significant impact on the development of dry eye disease in people with type 2 diabetes.42“
- SP is known to induce pro-inflammatory effects through autocrine or paracrine action on the immune cells in various ocular surface pathologies, including DED, as well. (Singh RB et al: Modulating the tachykinin: Role of substance P and neurokinin receptor expression in ocular surface disorders. Ocul Surf. 2022 July ; 25: 142–153. doi:10.1016/j.jtos.2022.06.007.)
Response
Thank you for the suggestion. We have included this statement in our introduction to enrich our statement of the problem. This is on lines 63-65 “Substance P is involved in neuro immunoregulation and maintenance of ocular surface comfort and homeostasis.5, 6, 9-13 Substance P is known to induce pro-inflammatory effects through autocrine or paracrine action on the immune cells in various ocular surface pathologies.14”
- Is there any data on whether the 2 study groups differ from each other and from age-matched healthy controls in terms of dry eye symptoms?
Response
Thank you for the comment. We have data on dry eye symptoms as assessed by the OSDI and this was included in Table 1. We have included this in the method section to make it obvious.
- A possible explanation for the non-significant differences in tear film substance P concentration between the two study groups could be dry eye disease.
Response
Thank you for the comment. There were no differences in dry eye symptoms between the two groups hence we do not believe dry eye disease may have influenced results.

Reviewer 2 Report
This study investigated the effect of chronic kidney disease (CKD) on tear film substance P concentration in type 2 diabetes patients, with participants divided into two groups - those with concurrent CKD and those without. A series of tests and assays were performed to measure the tear film substance P concentration and evaluate corneal nerve fiber density and length. However, the results demonstrated no significant difference in tear film substance P concentration between the groups, regardless of CKD presence or the severity of peripheral neuropathy. However, it was observed that corneal nerve fiber density and length were significantly reduced in patients with concurrent CKD, suggesting that corneal nerve parameters could serve as better indicators for detecting neuropathy and nephropathy-related corneal nerve changes in type 2 diabetes patients.
However, several critical concerns were identified that hinder the quality and credibility of the study. These concerns necessitate substantial revisions to the manuscript.
· The authors have based their study on a small sample size of 50 participants (25 in each group). Given the high prevalence of Type 2 Diabetes and CKD, a larger sample size would significantly enhance the robustness of the findings.
· The authors fail to describe the inclusion and exclusion criteria for the selection of participants in this study. Lack of information regarding the age, sex, disease duration, and other demographic factors of participants questions the generalizability of the results.
· The authors have used multiple statistical tests, but the choice of these tests is not justified. Moreover, the study lacks control for potential confounding variables that may affect the outcomes, including age, sex, or the presence of other comorbidities.
· The absence of a control group (i.e., individuals without Type 2 diabetes) is a significant omission in the experimental design. Without a control group, it is challenging to ascertain if the observed changes are due to Type 2 diabetes, CKD, or a combination of both.
· The authors state that there is no significant difference in tear film substance P concentration between T2DM-CKD and T2DM-no CKD groups, which contradicts the title and the initial hypothesis of the study. The link between tear film substance P concentration and corneal nerve parameters is unclear, which makes the paper's conclusions rather ambiguous.
· The authors' interpretation of their results is not convincing. They have also not adequately discussed their findings in light of existing literature. The relevance of their study to the larger clinical and scientific community is unclear.
· The manuscript does not sufficiently explain the biological mechanisms that might link CKD, Type 2 diabetes, corneal nerve loss, and changes in tear film substance P concentration. More in-depth discussion and exploration of the underlying biology are required to strengthen the study.
In its current form, the manuscript falls short of providing compelling evidence for the stated objectives. Thorough revisions addressing the above concerns are necessary to improve the manuscript. The authors are encouraged to redesign the study with a larger sample size, better-defined participant selection criteria, and inclusion of a control group. Additionally, a more detailed and clear presentation of the data, as well as comprehensive discussion of the results in the context of existing literature, is essential.
Author Response
Reviewer 2
This study investigated the effect of chronic kidney disease (CKD) on tear film substance P concentration in type 2 diabetes patients, with participants divided into two groups - those with concurrent CKD and those without. A series of tests and assays were performed to measure the tear film substance P concentration and evaluate corneal nerve fiber density and length. However, the results demonstrated no significant difference in tear film substance P concentration between the groups, regardless of CKD presence or the severity of peripheral neuropathy. However, it was observed that corneal nerve fiber density and length were significantly reduced in patients with concurrent CKD, suggesting that corneal nerve parameters could serve as better indicators for detecting neuropathy and nephropathy-related corneal nerve changes in type 2 diabetes patients.
Response
Thank you.
However, several critical concerns were identified that hinder the quality and credibility of the study. These concerns necessitate substantial revisions to the manuscript.
Response
Thank you very much for this careful review. We have considered and carefully addressed these critical concerns and outlined our responses below.
- 1. The authors have based their study on a small sample size of 50 participants (25 in each group). Given the high prevalence of Type 2 Diabetes and CKD, a larger sample size would significantly enhance the robustness of the findings.
Response
Thank you for your comment. We did a sample size calculation which revealed a minimum of 23 participants was adequate to detect a significant difference based on parameters from a study that demonstrated a difference in tear film substance P concentration between a diabetes group and a control group. Hence, 25 participants in each group matched for age, sex, BMI, HbAIc and Total Neuropathy Score should be adequate. However, we do agree with the reviewer that a larger sample size is always desirable and may improve the robustness of the study findings hence we added it to the limitations of the study.
- The authors fail to describe the inclusion and exclusion criteria for the selection of participants in this study. Lack of information regarding the age, sex, disease duration, and other demographic factors of participants questions the generalizability of the results.
Response
Thank you for the comment. We have included inclusion and exclusion criteria as requested. Concerning information on age, sex, disease duration and other demographic factors they were clearly presented in Table 1. We kindly request the reviewer to see this information in Table 1 as attached.
“Only participants meeting the following criteria were included in the study:
- All participants included in the study were aged above 18 years and willing to provide written informed consent.
- Only people with type 2 diabetes were recruited.
Participants were excluded if they had a history of any disease known to cause neuropathy such as vitamin B12 deficiency, use of chemotherapy or immunosuppressive medications, corneal abrasion, corneal ectasia, allergies to anaesthetic eye drops, usage of any topical steroidal and non-steroidal anti-inflammatories, current ocular infection, were contact lens wearers or had undergone cataract surgery in the last 6 months.”
Table 1: Clinical and demographic characteristics
|
Parameter |
T2DM-CKD (n = 25)
|
T2DM-no CKD (n = 25)
|
P-value |
|
|
Age, years |
70.8 ± 8.5 |
68.2 ± 8.5 |
P = 0.29 |
|
|
Sex, % Male |
64 |
72 |
P = 0.54 |
|
|
Body mass index, kg/m2 |
31.7 ± 6.9 |
32.1 ± 6.7 |
P = 0.98 |
|
|
Duration of diagnosis, years |
20.7 ± 8.8 |
14.7 ± 12.5 |
P = 0.06 |
|
|
HbA1c, % |
8.1 ± 1.8 |
8.7 ± 2.1 |
P = 0.25 |
|
|
Serum Urea, mg/dL |
10.8 ± 4.6 |
6.4 ± 1.9 |
P < 0.001 |
|
|
Creatinine, mg/dL |
171.2 ±118.5 |
76.3 ± 16.1 |
P < 0.001 |
|
|
Estimated glomerular filtration rate, mL/min/1.73 m2 |
41.3 ± 19.2 |
80.4 ± 10.9 |
P < 0.001 |
|
|
Urine ACR, mg/mmol |
44.3 ± 88.1 |
3.8 ±3.9 |
P =0.03 |
|
|
Serum Potassium, mmol/l |
4.5 ± 0.3 |
4.3 ± 0.5 |
P = 0.04 |
|
|
Total cholesterol, mmol/L |
3.8 ± 1.1 |
3.8 ± 1.0 |
P = 0.71 |
|
|
High density lipoprotein, mmol/L |
1.1 ± 0.4 |
1.3 ± 0.4 |
P = 0.24 |
|
|
Low density Lipoprotein, mmol/L |
1.8 ± 0.8 |
1.7 ± 0.9 |
P = 0.80 |
|
|
Triglycerides, mmol/L |
2.2 ± 2.1 |
1.7 ± 1.4 |
P = 0.40 |
|
|
Total Neuropathy Score (scores)
|
6.9 ± 5.4 |
6.4 ± 5.2 |
P = 0.74 |
|
|
Ocular surface staining (scores) |
3.6 ± 1.5 |
2.7 ± 2.3 |
P = 0.09 |
|
|
Ocular Surface Disease Index (scores) |
13.0 ± 11.4 |
12.6 ± 11.6 |
P = 0.76 |
|
|
Ocular Pain Assessment Survey (scores) |
3.5 ± 5.6 |
3.6 ± 4.9 |
P = 0.96 |
|
3.The authors have used multiple statistical tests, but the choice of these tests is not justified.
Response
Thank you for the comment. We modified our statistical analysis section and provided justification for the analysis done. This is on lines 192-210 which reads “Data analyses were conducted using SPSS version 23 (IBM Corp: Armonk, NY, USA) and Graph Pad Prism 9.0 (Graph Pad Software Inc., San Diego, CA, USA). Descriptive statistics were computed as percentages and counts for categorical variables and as means and standard deviations for continuous data. All the parameters were continuous variables except sex, which was categorical and analyzed with the χ2 test. Normality testing was done with the Shapiro–Wilk test and the visual inspection of the quantile-quantile and detrended plots. Data that showed normal distribution, such as the corneal nerve parameters, were analyzed using an independent sample t-test. For nonparametric data, such as tear film substance P concentration, a Mann–Whitney U test was used to determine any significant differences between the two diabetes groups. A P < 0.05 was considered statistically significant. Correlation analysis, Pearson r was performed to determine the relationship between tear film substance P concentration (logarithmically transformed) and ocular surface parameters as well as metabolic indicators. Finally, hierarchical multiple linear regression analysis controlling for age, duration of diagnosis and Total Neuropathy Score was conducted to determine whether corneal nerve parameters and kidney function test accounted for any significant variance in the logarithmically transformed tear film substance P concentration. Hierarchical multiple linear regression was conducted to control for the potential effects of duration of disease as the means of the two groups were quite different even though not reaching statistical significance.”
- Moreover, the study lacks control for potential confounding variables that may affect the outcomes, including age, sex, or the presence of other comorbidities.
Response
Thank you for your comment. Age, sex and presence of other co-morbidities were matched between the groups. From Table 1 it can be clearly seen that lipid profile, BMI, HbA1c and Total Neuropathy Scores were similar between the two groups. Only “duration of disease” had a marked difference in means although not reaching statistical significance. We nevertheless controlled for its effects in hierarchical multiple linear regression analysis.
- The absence of a control group (i.e., individuals without Type 2 diabetes) is a significant omission in the experimental design. Without a control group, it is challenging to ascertain if the observed changes are due to Type 2 diabetes, CKD, or a combination of both.
Response
Thank you for the comment. We agree with the reviewer that the inclusion of a control group would be desirable, and we have added it to the limitation of the study. However, previous studies in our group have already compared Type 2 with CKD and type 2 diabetes without CKD to healthy controls and whether Type 2 with CKD and type 2 diabetes without CKD groups were matched or unmatched for peripheral neuropathy, a greater corneal nerve loss (reduced corneal nerve fiber density and length) were observed compared to healthy controls. The previous studies are consistent with the current findings where type 2 diabetes with CKD had greater corneal loss compared to Type 2 diabetes without CKD.
- Tummanapalli SS, Issar T, Yan A, Kwai N, Poynten AM, Krishnan AV, Willcox MDP, Markoulli M. Corneal nerve fiber loss in diabetes with chronic kidney disease. Ocul Surf. 2020 Jan;18(1):178-185. doi: 10.1016/j.jtos.2019.11.010. Epub 2019 Nov 23. PMID: 31770601.
- Asiedu K, Markoulli M, Tummanapalli SS, Chiang JCB, Alotaibi S, Wang LL, Dhanapalaratnam R, Kwai N, Poynten A, Krishnan AV. Impact of Chronic Kidney Disease on Corneal Neuroimmune Features in Type 2 Diabetes. J Clin Med. 2022 Dec 20;12(1):16. doi: 10.3390/jcm12010016. PMID: 36614815; PMCID: PMC9820846
- The link between tear film substance P concentration and corneal nerve parameters is unclear, which makes the paper's conclusions rather ambiguous.
Response
Thank you for your comments. The corneal confocal microscopy measures comprising corneal nerve fiber density – (the total number of main nerves per square millimeter) and corneal nerve fiber length – (the total length of main nerves and nerve branches per square millimeter) which are primary indicators of cornea nerve loss. On the ocular surface, substance P is predominantly produced by the ophthalmic branch fibers of the trigeminal ganglion which innervate the cornea. The corneal nerves are the major source of tear film substance P hence corneal nerve degeneration or loss is expected to affect tear film substance P concentration. This is the case for type 1 diabetes. Exploring this in T2DM-CKD and T2DM-no CKD groups rather showed interesting findings of the two groups being similar for tear film substance P concentration despite greater corneal nerve loss in T2DM-CKD. This is captured on lines 54-61 “Recent studies have shown greater corneal nerve loss in diabetic chronic kidney disease compared to type 2 diabetes alone when peripheral neuropathy is matched between the two groups.4 Substance P in the tear film is mainly released by corneal neurons and exerts its physiological effects through the neurokinin receptors on the ocular surface.5, 6 Hence, the hypothesis explored in the present study is that reductions in corneal nerve fibers because of diabetic chronic kidney disease 4, 7, 8 may be associated with an alteration in the concentrations of substance P in tears. Measuring tear film neuropeptides such as substance P concentration may assist in detecting subclinical neuropathy in diabetic chronic kidney disease.”
- The authors state that there is no significant difference in tear film substance P concentration between T2DM-CKD and T2DM-no CKD groups, which contradicts the title and the initial hypothesis of the study.
Response
Thank you very much for the comment. We have modified our title to “CHRONIC KIDNEY DISEASE HAS NO IMPACT ON TEAR FILM SUBSTANCE P CONCENTRATION IN TYPE 2 DIABETES” and modified our hypothesis for it to read better “Hence, the hypothesis explored in the present study is that reductions in corneal nerve fibers because of diabetic chronic kidney disease 4, 7, 8 may be associated with an alteration in the concentrations of substance P in tears. Measuring tear film neuropeptides such as substance P concentration may assist in detecting subclinical neuropathy in diabetic chronic kidney disease”.
8.The authors' interpretation of their results is not convincing. They have also not adequately discussed their findings in light of existing literature. The relevance of their study to the larger clinical and scientific community is unclear.
Response
Thank you for your comment. We have added a few paragraphs explaining our results in the context of dry eye or ocular surface discomfort as suggested by reviewer 1. We have also refined our discussion to include the limitations of the study and refined the previous discussion. Two new paragraphs are captured on lines 275-301 which reads “Longstanding hyperglycemia in type 2 is a major cause of corneal nerve loss and peripheral nerve damage; resulting from advanced glycation end products, disruptions in nerve excitability or signaling cascades, uncontrolled generation of reactive oxygen species , and abnormal hemodynamic regulatory systems.34, 35 Concurrent chronic kidney disease in type 2 diabetes worsens these abnormalities36, 37 facilitating a greater corneal nerve loss. Furthermore, chronic kidney disease is closely associated with hyperkalaemia which mediate nerve dysfunction in chronic kidney disease by altering nerve ion channel function.38, 39 Nerve dysfunction is closely associated with the reduction in sodium–potassium pump activity common in type 2 diabetes.40 Reduced sodium–potassium pump activity and hyperkalaemia may combine synergistically to impair corneal nerves and other peripheral nerves more significantly in diabetic chronic kidney disease than type 2 diabetes alone.
Dry eye disease is very common in people with type 2 diabetes.41 Ocular surface inflammation is central to the pathogenesis of dry eye disease and ocular surface discomfort.42 Inflammation has a significant impact on the development of dry eye disease in people with type 2 diabetes.42 In laser-assisted in situ keratomileusis (LASIK), corneal nerve loss is accompanied by more severe ocular surface discomfort43, 44; however, corneal nerve loss in LASIK is distinctively different from type 2 diabetes. In LASIK, the corneal nerve loss is rapid, and it is accompanied by an increased level of tear film substance P concentration45, which enhances neuroinflammatory pathways; however, in type 2 diabetes, there is gradual loss of corneal nerve fibers, and no difference in tear film substance P concentration is observed between people with type 2 diabetes and healthy controls.15, 46The introduction of chronic kidney disease represents a condition that leads to severer and faster progression of neuropathy implying that additional corneal nerve loss in diabetic chronic kidney disease may alter tear film substance P concentration. However, no differences in tear film substance P concentration were observed between the two groups potentially explaining why there were no differences in ocular surface discomfort and ocular pain between the two groups in the current study and a previous study.7”
9.The manuscript does not sufficiently explain the biological mechanisms that might link CKD, Type 2 diabetes, corneal nerve loss, and changes in tear film substance P concentration. More in-depth discussion and exploration of the underlying biology are required to strengthen the study.
Response
Thank you for the comment. We have provided some biological mechanisms that link chronic kidney disease, type 2 diabetes, corneal nerve loss and changes in tear film substance P concentration. This is captured on line 275-285 which reads “Longstanding hyperglycemia in type 2 is a major cause of corneal nerve loss and peripheral nerve damage; resulting from advanced glycation end products, disruptions in nerve excitability or signaling cascades, uncontrolled generation of reactive oxygen species , and abnormal hemodynamic regulatory systems.34, 35 Concurrent chronic kidney disease in type 2 diabetes worsens these abnormalities36, 37 facilitating a greater corneal nerve loss. Furthermore, chronic kidney disease is closely associated with hyperkalaemia which mediate nerve dysfunction in chronic kidney disease by altering nerve ion channel function.38, 39 Nerve dysfunction is closely associated with the reduction in sodium–potassium pump activity common in type 2 diabetes.40 Reduced sodium–potassium pump activity and hyperkalaemia may combine synergistically to impair corneal nerves and other peripheral nerves more significantly in diabetic chronic kidney disease than type 2 diabetes alone.7”
In its current form, the manuscript falls short of providing compelling evidence for the stated objectives. Thorough revisions addressing the above concerns are necessary to improve the manuscript. The authors are encouraged to redesign the study with a larger sample size, better-defined participant selection criteria, and inclusion of a control group. Additionally, a more detailed and clear presentation of the data, as well as comprehensive discussion of the results in the context of existing literature, is essential.
Response
Thank you. The authors have provided the inclusion criteria for the study, clarified the sample size, provided a comprehensive discussion of the results in the context of current literature, and provided limitations of the study.

Round 2
Reviewer 2 Report
The manuscript was revised well and the quality has been improved. I recommend to accept.